# Soil Sulfur Sources Differentially Enhance Cadmium Tolerance in Indian Mustard (*Brassica juncea* L.)

Iqbal R. Mir [1], Bilal A. Rather [1], Asim Masood [1], Arif Majid [1], Zebus Sehar [1], Naser A. Anjum [1], Adriano Sofo [2,*], Ilaria D'Ippolito [2] and Nafees A. Khan [1,*]

1. Plant Physiology and Biochemistry Laboratory, Department of Botany, Aligarh Muslim University, Aligarh 202002, India; m3riqbal@gmail.com (I.R.M.); saffibilal@gmail.com (B.A.R.); asim.bot@gmail.com (A.M.); arifmajid8@gmail.com (A.M.); seharzebus5779@gmail.com (Z.S.); dnaanjum@gmail.com (N.A.A.)
2. Department of European and Mediterranean Cultures: Architecture, Environment, Cultural Heritage (DiCEM), University of Basilicata, 75100 Matera, Italy; dippolito.ilaria@libero.it
* Correspondence: adriano.sofo@unibas.it (A.S.); naf9.amu@gmail.com (N.A.K.)

**Abstract:** The effect of four soil-applied sulfur (100 mg S kg$^{-1}$ soil (100S) and 200 mg S kg$^{-1}$ soil (200S)) in different sources (elemental S, ammonium sulfate, gypsum or magnesium sulfate) in protecting mustard (*Brassica juncea* L. (Czern & Coss.)) from cadmium effects was studied. Based on the observed reduction in growth and photosynthesis in plants subjected to 100 and 200 mg Cd kg$^{-1}$ soil, *B. juncea* cv. Giriraj was selected as the most Cd-tolerant among five cultivars (namely, Giriraj, RH-0749, Pusa Agrani, RH-406, and Pusa Tarak). Sulfur applied to soil mitigated the negative impact of Cd on sulfur assimilation, cell viability, and photosynthetic functions, with a lower lipid peroxidation, electrolyte leakage, and contents of reactive oxygen species (ROS: hydrogen peroxide, $H_2O_2$, and superoxide anion, $O_2^{\bullet-}$). Generally, added S caused higher activity of antioxidant enzymes (ascorbate peroxidase, catalase and superoxide dismutase), contents of ascorbate (AsA) and reduced glutathione (GSH); increases in the activities of their regenerating enzymes (dehydroascorbate reductase and glutathione reductase); as well as rises in S assimilation, biosynthesis of non-protein thiols (NPTs), and phytochelatins (PCs). Compared to the other S-sources tested, elemental S more prominently protected *B. juncea* cv. Giriraj against Cd-impacts by minimizing Cd-accumulation and its root-to-shoot translocation; decreasing cellular ROS and membrane damage, and improving Cd-chelation (NPTs and PCs), so strengthening the defense machinery against Cd. The results suggest the use of elemental S for favoring the growth and development of cultivated plants also in Cd-contaminated agricultural soils.

**Keywords:** antioxidants; ascorbate; *Brassica juncea*; cadmium stress; Cd defense and tolerance; glutathione; Indian mustard; sulfur assimilation

## 1. Introduction

The issues concerning the continuous accumulation of metals in agricultural soils because of the anthropogenic activities have been widely reviewed and discussed over the past 30 years [1–3]. Particularly, cadmium (Cd), a hazardous heavy metal pollutant, has wide distribution, short half-life, and higher solubility in water, without a known biological function in plants. Cadmium enters the human food chain by getting accumulated in agricultural crops, thus causing severe threats to human and animal health [4].

In nature, Cd is accumulated in soil primarily through phosphatic fertilizers, irrigation with polluted water, and weathering of parent material, and also through volcanic eruptions [5]. The presence of Cd potentially stimulates the major plant enzymes, like NADPH oxidases, whose activity produce reactive oxygen species (ROS), such as singlet oxygen ($^1O_2$), hydroxyl radicles ($\bullet$OH), superoxide anion ($O_2^{\bullet-}$), and hydrogen peroxide

($H_2O_2$). These, in turn, cause membrane lipid peroxidation, and thus disturb redox homeostasis [5,6]. Plants with symptoms of Cd toxicity exhibit reduced chlorophyll content, plant biomass, shoot growth, number of flowers and fruits, and plant yield [7,8]. The photosynthetic efficiency of Cd-exposed plants may vary from low to high inhibition, and it differs among the plant species and the dose of Cd experienced. The Cd-inhibited alterations include changes in the chloroplast membranes and photosystems (PS) I and II, the inhibition of various cytosolic enzymes and of the enzymes of Calvin-Benson cycle [5,9], and the disturbance in the sulfur (S), nitrogen (N) and carbohydrate metabolism [10,11]. To counter Cd-induced phytotoxicity and excessive ROS, plants employ a series of defense mechanisms, comprising enzymatic antioxidants (such as superoxide dismutase, SOD; ascorbate peroxidase, APX; catalase, CAT; and glutathione reductase, GR), non-enzymatic antioxidants (such as ascorbate, AsA; glutathione, GSH), and non-protein thiols (NPTs).

Plant's inherent capacity of counteracting the potential impact of Cd has been widely reported to be strengthened with supplying plants with mineral nutrients, such as sulphur (S). This latter is the fourth essential macronutrient and is an integral component of the amino acids cysteine (Cys) and methionine (Met), antioxidants (GSH), heavy metal chelators (PCs, metallothioneins), prosthetic groups, co-enzymes, vitamins, secondary metabolites, thioredoxin system, and sulfolipids [12]. Plants absorb S via two mechanisms, one from the soil by sulfate transporters (energy-dependent process) and other via air sulfur dioxide ($SO_2$) through stomata. Then, S enters metabolism by getting converted into sulfate or S-containing amino acids [13]. During S-assimilation, by the action of enzymes such as ATP sulfurylase (ATP-S) and O-acetylserine (thiol) lyase (OASTL), sulfate is incorporated into Cys as the first product of S assimilation, which acts as a source of reduced S [14]. GSH is a tripeptide consisting of glycine (Gly), glutamate (Glu), and Cys, whereas PCs are peptides with repeated units of $\gamma$-glutamyl cysteine [15]. It is well established that GSH molecules are the substrate for the synthesis of PCs [16] and that Cys acts as a precursor for GSH and its derivatives [17]. Moreover, the activity of enzymatic antioxidants increases in Cd-exposed plants after S supplementation, which lowers $H_2O_2$ content, membrane peroxidation, and electrolyte leakage [12,18]. Sulfur is also known to restrain Cd transport to shoot and reduce Cd accumulation by promoting the synthesis of non-protein thiols (NPTs) pool (including PCs and GSH) [19]. Furthermore, reduced glutathione (GSH) was reported to scavenge excess ROS via AsA-GSH cycle and reduce Cd-induced phytotoxicity [17].

Plants belonging to *Brassica* genus have comparatively higher S demand owing to synthesis of sulfur compounds like glucosinolates, and hence are more sensitive to S deficiency [15,16,20]. Mustard (*Brassica juncea* L. (Czern & Coss.)) has been one of the most studied crop plants for its ability to extract heavy metals, including Cd [7,20,21]. Being a potential hyperaccumulator of heavy metals, it is capable of accumulating a high concentration of Cd in both roots and aboveground parts. However, Cd toxicity responses and detoxification mechanisms are genotype-dependent, and different cultivars respond distinctly because of their varying genetic potential to Cd stress [20]. The smart selection of plant cultivars, with the potential to resist Cd-induced phytotoxicity, could be the best strategy to counter the inhibitory effects of Cd in mustard plants. Moreover, the identification of the most suitable S sources for growth and development of plants under Cd stress could help to amend agricultural soil and enhance plant survival. With these postulations, the present study was conducted (a) to assess the effectiveness of S sources for the alleviation of Cd-induced phytotoxicity in mustard plants, and (b) to investigate the physiological and biochemical mechanisms involved in their photosynthetic performance and growth dynamics in the same plants.

## 2. Materials and Methods

### 2.1. Plant Material, Growth Conditions and Experimental Layout

Healthy uniform seeds of five mustard [*Brassica juncea* L. (Czern & Coss.)] cultivars, Giriraj, RH-0749, Pusa Agrani, RH-406, and Pusa Tarak were obtained from the Indian Agricultural Research Institute, New Delhi. Seeds were surface sterilized with 0.4% (*v/v*)

sodium hypochlorite (NaClO) and then rinsed repetitively with double distilled water. Surface-sterilized seeds were sown in 23-cm diameter pots filled with 5 kg of reconstituted soil with peat:compost:sand (4:1:1, *w/w*). The pots used in the experiments had holes, however, the plants were watered by calculating field capacity, so that there was minimum leaching. Before sowing, soil samples were collected randomly from different pots for selected soil-characteristics analysis. Soil texture; pH in water; electrical conductivity; and S, N, P, and K contents were measured by soil standard methods, according to Pansu and Gautheyrou [22]; soil Cd content was determined by atomic absorption spectrophotometry (GBC, 932 Plus; GBC Scientific instruments, Braeside, Australia) (Table 1). The pots were kept in the greenhouse with day/night temperatures of $23/16 \pm 2$ °C, a 16/8 h light/dark period, and relative humidity of $60 \pm 4\%$.

**Table 1.** Mean values (*n* = 4) of physicochemical parameters of soils from experiments 1–3. nd, not detected.

| Soil Parameter | Unit of Measure | Experiment 1 | Experiment 2 | Experiment 3 |
|---|---|---|---|---|
| Texture | | Sandy loam | Sandy loam | Sandy loam |
| pH | | 7.83 | 7.64 | 7.78 |
| Electrical conductivity | $(ds\ m^{-1})$ | 0.48 | 0.51 | 0.43 |
| S | $(mg\ kg^{-1}\ soil)$ | 31.56 | 29.85 | 26.17 |
| N | $(mg\ kg^{-1}\ soil)$ | 72.51 | 76.94 | 78.68 |
| P | $(mg\ kg^{-1}\ soil)$ | 8.32 | 9.79 | 8.11 |
| K | $(mg\ kg^{-1}\ soil)$ | 115.64 | 133.81 | 138.65 |
| Cd | $(mg\ kg^{-1}\ soil)$ | nd | nd | nd |

In the first experiment, five *Brassica* cultivars were screened for their tolerance to different Cd levels (0, 100 and 200 mg Cd kg$^{-1}$ of soil), based on growth and photosynthetic attributes. The source of Cd was cadmium chloride ($CdCl_2$), and it was added to pots at the time of sowing.

The second experiment was conducted on *B. juncea* cultivar Giriraj. The plants were grown with 100 mg S kg$^{-1}$ soil (100S) and 200 mg S kg$^{-1}$ soil (200S). Sulfur was supplemented as elemental S ($S^0$), ammonium sulfate [$(NH_4)_2SO_4$], gypsum ($CaSO_4 \cdot 2H_2O$) or magnesium sulfate ($MgSO_4$). As 200S improved photosynthesis, growth, and S-assimilation higher than 100S, it was selected for another experiment.

In the third experiment, plants were grown either with 200 mg Cd kg$^{-1}$ soil, $S^0$, [$(NH_4)_2SO_4$], $CaSO_4 \cdot 2H_2O$ or $MgSO_4$, or in combined treatment of Cd + $S^0$, Cd + $MgSO_4$, Cd + $CaSO_4 \cdot 2H_2O$, and Cd + $MgSO_4$. A control group of plants without sulfur was maintained. Elemental S was given 15 days before sowing, while all the other sources were supplied at the time of sowing.

Treatments in all the experiments were arranged in a factorial randomized block design, and the number of replicates for each treatment was four (*n* = 4). At 30 days after sowing (DAS), analyses were made to study gas exchange, photosynthetic efficiency, growth characteristics, oxidative stress, antioxidant levels ad activity, S and N metabolism, and antioxidant system.

*2.2. Growth Parameters*

Growth attributes were measured at 30 DAS for experiments 1, 2, and 3. Plants were uprooted carefully from the pots and washed to remove the soil and dust from roots. Then, plants were weighed to determine the fresh biomass and were later kept for drying in an oven at 80 °C to record dry biomass. Leaf area was measured using leaf area meter (LA211, Systronics, New Delhi, India).

*2.3. Estimation of Cadmium and Sulfur Content*

The concentration of Cd was analyzed using atomic absorption spectrophotometer. Root and leaf samples were dried in an oven for 2 days at 80 °C. The dried tissue was

weighed, ground to powder, and the resulting powder was digested with concentrated $HNO_3$:$HClO_4$ (3:1; *v/v*). Cd content was determined by atomic absorption spectrophotometry (GBC, 932 Plus; GBC Scientific Instruments, Braeside, Australia). Cd translocation factor (TF) was calculated as shoot to root ratio of Cd concentration [4]. Tolerance index (TI) was calculated as the ratio of dry weights of plants exposed to Cd to that under control conditions [20].

Sulfur content for experiment 3 was determined in oven-dried leaves (0.1 g) and roots (0.5 g) digested in a mixture of concentrated $HNO_3$ and 60% strength $HClO_4$ (85:15; *v/v*), using the turbidimetric method of Chesnin and Yien [23] with some modifications. A 5 mL reaction mixture was made up by adding 1.0 g $BaCl_2$ and 2.5 mL gum acacia, and the final volume was made up to 25 mL by adding distilled water and contents were thoroughly shaken to dissolve completely $BaCl_2$. The values were recorded spectrophotometrically at 415 nm from the time of turbidity development, and a blank was run after determining each set simultaneously.

### 2.4. Gas Exchange and Photosynthetic Parameters

Net photosynthesis ($P_N$), stomatal conductance ($g_s$), and intercellular $CO_2$ concentration ($C_i$) for all the three experiments were measured in fully expanded and intact topmost leaves in various treatments and replicates using an infrared gas analyser (CID-340, Photosynthesis system, Bio-Science, Washington, DC, USA). The measurements were made on a sunny day between 10:00 and 11:30 a.m. at photosynthetic active radiation (PAR) of 720 $\mu$mol m$^{-2}$ s$^{-1}$ and atmospheric $CO_2$ concentration of 415 $\mu$L L$^{-1}$. Chlorophyll content (SPAD values) was measured in fully expanded uppermost leaves with a SPAD chlorophyll meter (502 DL PLUS, Spectrum Technologies, Aurora, IL, USA).

Maximal PSII photochemical efficiency ($F_v/F_m$) of intact leaf from the top for experiments 2 and 3 was measured using a chlorophyll fluorometer (Junior Pam, Heinz Walz, Germany). Leaf samples were kept in the dark for 30 min to measure maximum fluorescence ($F_m$) and minimum fluorescence ($F_o$). The value of $F_o$ was determined from a weak pulse (0.1 $\mu$mol m$^{-2}$ s$^{-1}$), while Fm was obtained from saturating pulse (>6000 $\mu$mol m$^{-2}$ s$^{-1}$). Variable fluorescence ($F_v$) was calculated by subtracting $F_o$ from $F_m$, and $F_v/F_m$ ratio was calculated, which is a measure of maximum quantum yield efficiency of PSII. The activity of ribulose-1,5-bisphosphate carboxylase/oxygenase (Rubisco) was ascertained adopting the method by Usuda [24] by monitoring the oxidation of NADH at 30 °C at 340 nm. The activity was measured after the addition of enzyme extract followed by 0.2 mM ribulose-1,5-bisphosphate (RuBP). The detailed procedure is reported in Rather et al. [2].

### 2.5. Assessment of Oxidative Damage

The content of $O_2^{\bullet-}$ and $H_2O_2$ was determined spectrophotometrically by adopting the method of Wu et al. [25] and Okuda et al. [26], respectively. The degree of production of $O_2^{\bullet-}$ and $H_2O_2$ in vivo was validated via the histochemical method of Kumar et al. [27]. The electrolyte leakage (EC) and lipid peroxidation by thiobarbituric acid reactive substances (TBARS) were measured spectrophometrically according to Sullivan and Ross [28] and Dhindsa et al. [29], respectively.

### 2.6. Antioxidant Enzymes and Non-Enzymatic Antioxidants

Fresh leaf tissues (0.5 g) were homogenized in chilled mortar and pestle with an extraction buffer containing 0.05% (*v/v*) Triton X 100 and 1% (*w/v*) polyvinylpyrrolidone in 100 mM potassium phosphate buffer (pH 7.0). The homogenate was centrifuged at 15,000× *g* for 20 min at 4 °C. The activity of SOD (EC 1.15.1.1) was accessed according to the method of Giannopolitis and Ries [30] by observing the inhibition of photochemical reduction of nitro blue tetrazolium (NBT). One unit of SOD activity was the amount of enzyme that inhibits the NBT reduction by 50% followed at 560 nm. CAT (1.11.1.6) activity was determined by the method provided by Aebi [31] by visualizing the disappearance of $H_2O_2$ at 240 nm. One unit of CAT activity was the amount that decomposes 1 $\mu$mol

of $H_2O_2$ min$^{-1}$ at 25 °C. The activity of APX (EC 1.11.1.11) was measured according to the method of Nakano and Asada [32] by observing a decrease in absorbance of ascorbate at 290 nm. One unit of AP activity was defined as the amount required to decompose 1 µmol of substrate min$^{-1}$ at 25 °C. GR (EC 1.6.4.2) activity was determined following the protocol of Foyer and Halliwell [33] with some modifications. Reaction mixture consisted of 0.5 mM oxidized GSH, 0.2 mM NADPH, phosphate buffer (25 mM, pH 7.8). One unit of GR activity was the amount necessary to decompose 1.0 µmol of NADPH min$^{-1}$ at 25 °C. Dehydroascorbate reductase (DHAR, EC 1.8.5.1) activity was assayed following the method of Pinto et al. [34]. A 0.01 increase in absorbance at 265 nm was defined as one unit of DHAR activity.

Reduced glutathione (GSH) was determined spectrophotometrically at 412 nm following the method of Griffith [35]. Reduced ascorbate (AsA) content was estimated using 2,6-dichlorophenol-indophenol-based titration method provided by Lu et al. [12].

### 2.7. Non-Protein Thiols and Total Phytochelatins Content

The content of non-protein thiols (NPTs) and total phytochelatins (PCs) was measured following the procedure of Lou et al. [36] with some modifications. NPT was extracted by homogenizing 0.2 g of leaf samples in 2 mL of 5% sulfosalicylic acid and then centrifuged at 10,000× $g$ for 15 min at 4 °C. For NPT determination, the reaction mixture contained 0.2 mL of the supernatant, 0.15 mL of 10 mM 5,5′dithiobis [2-nitrobenzoic acid] (DTNB), and 0.2 M Tris-HCl (pH 8.2). The reaction mixture was incubated for 20 min, and absorbance was measured spectrophotometrically at 412 nm. PCs content was calculated by subtracting total GSH content from the total amount of NPTs.

### 2.8. S-Assimilating Enzymes and S-Containing Amino Acids

The activity of ATP-S (EC 2.7.7.4) for experiments 2 and 3 was determined according to Lappartient and Touraine [37] by following spectrophotometrically at 340 nm the molybdate-dependent formation of pyrophosphate. The activity of OASTL (EC 4.2.99.8) was determined spectrophotometrically with the method of Riemenschneider et al. [38].

Leaf cysteine (Cys) content for experiments 2 and 3 was determined spectrophotometrically at 580 nm using the method of Gaitonde [39], with some modifications. In brief, 500 mg fresh leaves were homogenised in ice-cold perchloric acid (5%; $w/v$). The obtained suspension was centrifuged at 2800× $g$ for 1 h at 5 °C. Supernatant was filtered through Whatman No.1 filter paper. The content of methionine (Met) was determined according to Khan et al. [11].

### 2.9. Confocal Laser Microscopy to Study Root Cell Viability

Clean and thin sections of roots were dipped in 25 µM propidium iodide (PI) solution to visualize cell viability. After washing properly, root samples were analysed with a confocal microscope. The stained samples were visualized under a confocal microscope (Olympus Fluoview TM-FV1000, Olympus Life Sciences, Tokyo, Japan) with a 63× oil immersion objective, at excitation of 400–490 nm, emission ≥ 520 nm. Fluoview FV10 software, ver 1.7 (Olympus Life Sciences, Tokyo, Japan) was used to analyze and process the images. Analyses were performed on 30-day old roots.

### 2.10. Physiological Measurements of the Guard Cells

Fresh leaves from 30-day-old plants were plugged off from each branch of various treatments and were fixed by 2.5% glutaraldehyde, and stomatal images were captured by means of a Carl Zeiss EVO 40 scanning electron microscope (Zeiss, Aalen, Germany) at extra high tension and high voltage at 20 kV. Leaf samples were first fixed with 2.5% glutaraldehyde plus 2% paraformaldehyde ($v/v$) in 0.1 M phosphate buffer (pH 7.0) in equal quantity for 45 h, and then washed three times with phosphate buffer for 15 min at each step. The samples were then post fixed with 1% osmium oxide in phosphate buffer (pH 7.0) for 1 h and washed three times with the same phosphate buffer for 15 min.

Then, they were dehydrated by a graded series of ethanol (50, 70, 80, 90, 95, and 100%) for about 15–20 min at each step, and transferred to the mixture of alcohol and isoamyl acetate (1:1; *v/v*) for about 30 min. Finally, the samples were transferred to pure isoamyl acetate for 1 h and dehydrated with liquid $CO_2$. The dehydrated specimen was coated with gold-palladium and observed under the microscope. Analyses were performed on 30-day old leaves.

### 2.11. Statistical Analysis

Data were analysed statistically by analysis of variance (ANOVA) using SPSS v17.0 for Windows (IBM Corporation, New York, NY, USA). Least significant differences (LSDs) were calculated among treatments at $p < 0.05$ levels. More details are given in table and figure captions.

## 3. Results

### 3.1. Screening of Cultivars for Cd Tolerance

The effects of Cd on growth parameters (fresh plant biomass, dry plant biomass, leaf area) and photosynthetic attributes (chlorophyll content measured by SPAD, net photosynthesis, stomatal conductance) in the five cultivars of B. juncea were summarized in Table 2. Irrespective of the cultivars, both 100 Cd and 200 Cd levels impaired the tested photosynthetic and plant growth traits, compared to the control. However, the maximum adverse effect of Cd on plant photosynthetic and growth attributes was observed with 200 Cd, compared to 100 Cd. The highest Cd accumulation was recorded in roots and leaves of Pusa Tarak, and the lowest in Giriraj cultivar. Tolerance index (TI) of all the cultivars, calculated using data of plant dry mass treated with Cd 200, confirmed that Giriraj had the highest tolerance index (0.789) among all the tested cultivars, whereas Pusa Tarak was the most sensitive (Table 2). The cultivars showed tolerance to Cd in the following order: Giriraj > RH-0749 > Pusa Agrani > RH-406 > Pusa Tarak.

### 3.2. Response of Plants to Different S Sources and S Levels

To assess the S-requirement of the crop, we evaluated the effects of two S levels (100S and 200S) from four S sources (i.e., elemental S, $S^0$; ammonium sulfate, AS; gypsum, Gyp; and magnesium sulfate, MS) on growth and photosynthetic parameters, and S-assimilation (Table 3 and Figure 1). Plants showed differential responses to S sources, and the response was dose-dependent.

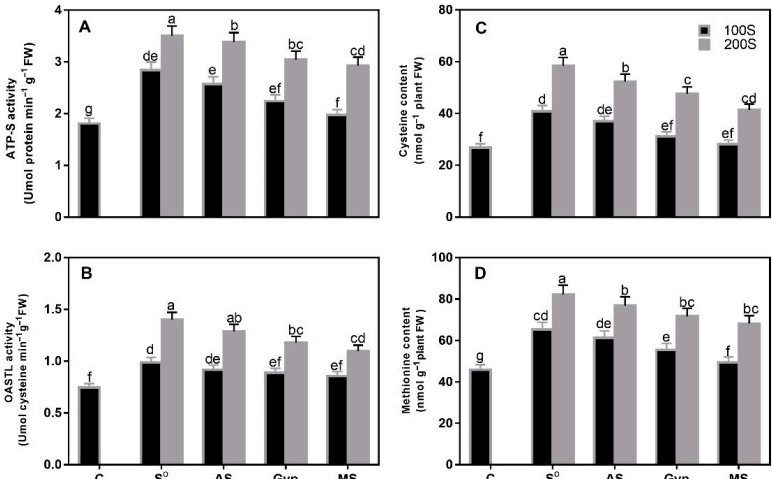

**Figure 1.** Foliar (**A**) Cysteine content, (**B**) Methonine content, (**C**) ATP-S activity, and (**D**) OASTL activity in *Brassica juncea* cv. Giriraj treated with elemental S ($S^0$), ammonium sulfate (AS), gypsum (Gyp) or magnesium sulfate (MS), under two S levels (100 and 200 mg S kg$^{-1}$). Data are presented as means $\pm$ SE (*n* = 4). Data with the same letter are not significantly different by LSD test at $p < 0.05$. ATP-S, ATP sulfurylase; C, control; FW, fresh weight; OASTL, O-acetylserine (thiol) lyase.

**Table 2.** Plant fresh biomass, plant dry biomass, chlorophyll content (SPAD values), net photosynthesis, stomatal conductance, intercellular $CO_2$, Cd content of roots and leaves, and tolerance index of five *Brassica juncea* cvs. Giriraj, RH-0749, Pusa Agrani, RH-406, and Pusa Tarak grown under three soil Cd levels [i.e., control 0 Cd (0 mg Cd kg$^{-1}$ soil), 100 Cd (100 mg Cd kg$^{-1}$ soil) and 200 Cd (200 mg Cd kg$^{-1}$ soil)]. Data are presented as means $\pm$ SE ($n = 4$). Data followed by same letter within columns are not significantly different by LSD test at $p < 0.05$. Cd, cadmium; DW, dry weight; nd, not detected.

| Cultivar | Cd Level | Plant Fresh Biomass | Plant Dry Biomass | Leaf Area | Chlorophyll Content | Net Photosynthesis | Stomatal Conductance | Cd Root Content | Cd Leaf Content | Tolerance Index |
|---|---|---|---|---|---|---|---|---|---|---|
| | (mg Cd kg$^{-1}$ Soil) | (g plant$^{-1}$) | | (cm$^2$ plant$^{-1}$) | | ($\mu$mol $CO_2$ m$^{-2}$ s$^{-1}$) | (mmol $H_2O$ m$^{-2}$ s$^{-1}$) | ($\mu$g g$^{-1}$ DW) | | |
| Giriraj | 0 (control) | 20.39 $\pm$ 0.51 [a] | 2.04 $\pm$ 0.10 [a] | 140.35 $\pm$ 7.06 [a] | 28.69 $\pm$ 1.44 [a] | 20.96 $\pm$ 1.05 [a] | 229.17 $\pm$ 11.53 [a] | nd | nd | |
| | 100 | 18.59 $\pm$ 0.47 [c,d] | 1.73 $\pm$ 0.09 [b,c] | 122.45 $\pm$ 6.16 [b] | 26.53 $\pm$ 1.34 [b,c] | 17.41 $\pm$ 0.88 [d,e] | 205.17 $\pm$ 10.33 [c,d] | 125.66 $\pm$ 6.32 [f] | 28.96 $\pm$ 1.46 [i] | |
| | 200 | 15.36 $\pm$ 0.39 [f] | 1.61 $\pm$ 0.08 [c] | 105.51 $\pm$ 5.31 [c,d] | 23.85 $\pm$ 1.20 [d] | 13.24 $\pm$ 0.67 [h] | 179.46 $\pm$ 9.03 [d] | 191.14 $\pm$ 9.62 [d,e] | 113.35 $\pm$ 5.71 [e] | 0.789 $\pm$ 0.041 [a] |
| RH-0749 | 0 (control) | 20.06 $\pm$ 0.50 [a] | 2.01 $\pm$ 0.10 [a] | 128.71 $\pm$ 6.45 [a,b] | 28.26 $\pm$ 1.42 [a] | 20.35 $\pm$ 1.02 [a] | 227.62 $\pm$ 11.46 [a,b] | nd | nd | |
| | 100 | 18.33 $\pm$ 0.46 [d,e] | 1.55 $\pm$ 0.08 [c,d] | 116.29 $\pm$ 5.85 [b,c] | 26.14 $\pm$ 1.32 [b,c] | 17.06 $\pm$ 0.86 [d,e] | 198.38 $\pm$ 9.98 [c,d] | 151.35 $\pm$ 0.62 [e,f] | 40.14 $\pm$ 2.02 [h] | |
| | 200 | 14.92 $\pm$ 0.38 [f] | 0.95 $\pm$ 0.05 [f] | 87.14 $\pm$ 4.39 [e] | 23.19 $\pm$ 1.17 [d,e] | 12.45 $\pm$ 0.63 [h,i] | 174.39 $\pm$ 8.78 [d,e] | 215.98 $\pm$ 10.87 [c] | 129.46 $\pm$ 6.52 [d] | 0.472 $\pm$ 0.026 [b] |
| Pusa Agrini | 0 (control) | 19.88 $\pm$ 0.50 [a,b] | 1.97 $\pm$ 0.10 [a,b] | 117.62 $\pm$ 5.92 [b,c] | 28.13 $\pm$ 1.42 [a,b] | 19.69 $\pm$ 0.99 [a,b] | 224.31 $\pm$ 11.29 [b,c] | nd | nd | |
| | 100 | 18.09 $\pm$ 0.46 [d,e] | 1.42 $\pm$ 0.07 [d] | 92.34 $\pm$ 4.65 [d,e] | 25.87 $\pm$ 1.30b [c] | 16.31 $\pm$ 0.82 [e,f] | 195.48 $\pm$ 9.84 [c,d] | 175.22 $\pm$ 8.82 [d,e] | 61.95 $\pm$ 3.12 [g] | |
| | 200 | 13.65 $\pm$ 0.34 [g] | 0.81 $\pm$ 0.04 [g] | 60.32 $\pm$ 3.04 [g,h] | 22.82 $\pm$ 1.15 [e] | 12.07 $\pm$ 0.61 [h,i] | 172.63 $\pm$ 8.69 [d,e] | 269.27 $\pm$ 13.55 [b] | 135.88 $\pm$ 6.84 [c] | 0.411 $\pm$ 0.035 [c] |
| RH-406 | 0 (control) | 19.64 $\pm$ 0.49 [b,c] | 1.94 $\pm$ 0.10 [a,b] | 106.76 $\pm$ 5.37 [c,d] | 28.02 $\pm$ 1.41 [a,b] | 19.11 $\pm$ 0.96 [b,c] | 223.44 $\pm$ 11.25 [b,c] | nd | nd | |
| | 100 | 17.83 $\pm$ 0.45 [e] | 1.36 $\pm$ 0.07 [d,e] | 84.81 $\pm$ 4.27 [e,f] | 25.44 $\pm$ 1.28 [c,d] | 15.94 $\pm$ 0.80 [f,g] | 193.56 $\pm$ 9.74 [c,d] | 188.65 $\pm$ 9.50 [d] | 76.54 $\pm$ 3.85 [f,g] | |
| | 200 | 12.67 $\pm$ 0.32 [g] | 0.74 $\pm$ 0.047 [g] | 52.74 $\pm$ 2.65 [h] | 20.46 $\pm$ 1.03 [f] | 11.35 $\pm$ 0.57 [i,j] | 166.13 $\pm$ 8.36 [e] | 295.84 $\pm$ 14.89 [b] | 142.69 $\pm$ 7.18 [b] | 0.381 $\pm$ 0.026 [d] |
| Pusa tarak | 0 (control) | 19.25 $\pm$ 0.48 [b,c] | 1.80 $\pm$ 0.09 [b] | 99.21 $\pm$ 4.99 [d,e] | 27.91 $\pm$ 1.40 [a,b] | 18.65 $\pm$ 0.94 [c,d] | 219.27 $\pm$ 11.04 [b,c] | nd | nd | |
| | 100 | 17.68 $\pm$ 0.44 [e] | 1.26 $\pm$ 0.06 [e] | 70.59 $\pm$ 3.55 [f,g] | 25.06 $\pm$ 1.26 [c,d] | 14.06 $\pm$ 0.71 [g,h] | 192.81 $\pm$ 9.70 [c,d] | 228.35 $\pm$ 11.49 [c] | 89.75 $\pm$ 4.52 [f] | |
| | 200 | 11.06 $\pm$ 0.28 [h] | 0.65 $\pm$ 0.03 [h] | 45.38 $\pm$ 2.28 [h] | 18.37 $\pm$ 0.92 [g] | 11.06 $\pm$ 0.56 [i,j] | 152.79 $\pm$ 79.20 [f] | 364.48 $\pm$ 18.35 [a] | 158.23 $\pm$ 7.96 [a] | 0.361 $\pm$ 0.023 [d] |

**Table 3.** Plant fresh biomass, plant dry biomass, chlorophyll content (SPAD values), net photosynthesis, stomatal conductance, intercellular $CO_2$ and maximal PSII efficiency ($F_v/F_m$ values) of *Brassica juncea* cv. Giriraj grown under three S levels [i.e., control 0S (0 mg S kg$^{-1}$ soil), 100S (100 mg S kg$^{-1}$ soil) and 200S (200 mg S kg$^{-1}$ soil)] and four S sources [i.e., elemental S (S$^0$), ammonium sulfate (AS), gypsum (Gyp) and magnesium sulfate (MS)]. Data are presented as means ± SE ($n = 4$). Data followed by same letter within columns are not significantly different by LSD test at $p < 0.05$. S, sulfur.

| S Source | S Level | Plant Fresh Biomass | Plant Dry Biomass | Chlorophyll Content | Net Photosynthesis | Stomatal Conductance | Intercellular $CO_2$ | Maximal PSII Efficiency |
|---|---|---|---|---|---|---|---|---|
| | (mg S kg$^{-1}$ Soil) | (g plant$^{-1}$) | | | ($\mu$mol $CO_2$ m$^{-2}$ s$^{-1}$) | (mmol $H_2O$ m$^{-2}$ s$^{-1}$) | ($\mu$mol $CO_2$ mol$^{-1}$) | |
| **Control** | 0 | 19.69 ± 0.83 [h] | 1.81 ± 0.08 [e] | 26.48 ± 1.56 [g] | 19.23 ± 1.13 [f] | 235.62 ± 11.11 [e] | 241.38 ± 10.15 [f] | 0.578 ± 0.03 [i] |
| **S$^0$** | 100 | 23.19 ± 1.07 [d] | 2.24 ± 0.11 [d] | 33.41 ± 1.97 [d] | 23.67 ± 1.39 [c] | 347.92 ± 15.19 [c] | 303.35 ± 12.56 [c,d] | 0.683 ± 0.03 [e] |
| | 200 | 26.19 ± 1.16 [a] | 3.76 ± 0.16 [a] | 39.45 ± 2.32 [a] | 26.98 ± 1.56 [a] | 430.49 ± 19.46 [a] | 394.66 ± 17.35 [a] | 0.814 ± 0.04 [a] |
| **AS** | 100 | 22.45 ± 0.99 [e] | 2.19 ± 0.10 [d] | 32.59 ± 1.92 [d,e] | 22.45 ± 1.32 [d] | 317.35 ± 12.80 [c,d] | 296.17 ± 11.90 [d,e] | 0.664 ± 0.03 [f] |
| | 200 | 25.64 ± 1.14 [b] | 3.19 ± 0.14 [b] | 37.62 ± 2.22 [b] | 25.17 ± 1.48 [b] | 410.39 ± 18.28 [a,b] | 371.76 ± 16.59 [a,b] | 0.785 ± 0.03 [b] |
| **Gyp** | 100 | 22.91 ± 1.01 [f] | 2.06 ± 0.10 [d,e] | 31.87 ± 1.88 [e,f] | 22.38 ± 1.30 [d] | 308.37 ± 13.45 [c,d] | 291.64 ± 11.28 [d,e] | 0.646 ± 0.03 [g] |
| | 200 | 25.31 ± 1.12 [b,c] | 3.02 ± 0.13 [b,c] | 36.44 ± 2.15 [b,c] | 25.03 ± 1.43 [b] | 396.45 ± 11.57 [b,c] | 330.51 ± 13.54 [b] | 0.769 ± 0.03 [c] |
| **MS** | 100 | 21.56 ± 1.01 [g] | 2.03 ± 0.09 [d,e] | 29.33 ± 1.73 [f] | 21.54 ± 1.26 [e] | 291.49 ± 11.98 [d,e] | 269.46 ± 10.57 [e] | 0.621 ± 0.03 [h] |
| | 200 | 24.76 ± 1.09 [c] | 2.88 ± 0.10 [cd] | 35.45 ± 2.09 [c,d] | 24.62 ± 1.41 [b,c] | 389.44 ± 17.04 [b,c] | 318.62 ± 13.05 [b,c] | 0.713 ± 0.03 [d] |

Photosynthetic and growth parameters were positively influenced by both S levels (100S and 200S). In contrast to 100S, the effects of 200S on growth and photosynthetic attributes were more conspicuous for all S sources. Among the four sources of S used, $S^0$ showed a more protruding effect than any other S source, both at 100 and 200S levels, and generally increased the values of growth and photosynthetic parameters. The entity of the effect of $S^0$ was followed by those of AS, Gyp, and MS. $S^0$-treated plants had increased chlorophyll content, net photosynthesis, intercellular $CO_2$ concentration, stomatal conductance and maximal PSII efficiency, plant fresh, and dry biomass, particularly at 200S. The increases in these characteristics were by 48.9, 37.7, 75.2, 70.9, 35.2, 40.1, and 86.3%, respectively, compared to the control (Table 3).

Of the two S levels, 200S was more efficacious in increasing the activity of ATP-S and OASTL by 93.3 and 87.8% ($S^0$), 86.7 and 73.0% (AS), 67.9 and 58.1% (Gyp), and 61.9 and 47.9% (MS), respectively, over the control (Figure 1A,B). A similar trend was observed for Cys and Met content, where 200S caused increased Cys content by two-times and Met content by 79.1%, in respect to the control (Figure 1C,D). The higher effectiveness of 200S improved plant growth more than 100S. For this reason, 200S treatment was considered for further study.

### 3.3. Effects of Different S Sources on Cd and S Accumulation

The accumulation of Cd in Cd-exposed plants was significantly higher in roots than leaves (Table 4). While all S sources reduced the accumulation of Cd in roots and leaves, $S^0$ was the most effective. $S^0$-treated plants recorded a reduction by 78.5% of Cd content in roots and by 84.3% in leaves, followed by AS, Gyp, and MS (Table 4). Moreover, plants treated with $S^0$ had the lowest value of translocation factor (TF = 0.590), when compared with the other S sources. The content of S was markedly reduced by 22.6% in roots and by 23.3% in leaves in Cd-stressed plants, compared to the control (Table 4). The accumulation of S in roots and leaves increased after the application of each S form, particularly for $S^0$ (Table 4).

**Table 4.** Effect of 200 mg S kg$^{-1}$ soil of elemental S ($S^0$), ammonium sulfate (AS), gypsum (Gyp) and magnesium sulfate (MS) on Cd content and S content in *Brassica juncea* cv. Giriraj under Cd stress (200 mg Cd kg$^{-1}$ soil). Data are presented as means $\pm$ SE ($n$ = 4). Data followed by same letter within columns are not significantly different by LSD test at $p < 0.05$. −Cd, without cadmium; +Cd, with cadmium; DW, dry weight; nd, not detected; S, sulfur; TF, translocation factor.

| Treatment | Cd Treatment | Cd Content | | S Content | | Cd TF |
|---|---|---|---|---|---|---|
| | | Roots | Leaves | Roots | Leaves | |
| | | (µg g$^{-1}$ DW) | | (mg g$^{-1}$ DW) | | |
| **Control** | −Cd | nd | nd | 4.07 ± 0.07 [e] | 4.29 ± 0.08 [g] | nd |
| **Cd** | +Cd | 166.96 ± 9.33 [a] | 138.59 ± 4.18 [a] | 3.12 ± 0.06 [f] | 3.32 ± 0.06 [h] | 0.830 [a] |
| **S₀** | −Cd | nd | nd | 6.77 ± 0.12 [a] | 7.98 ± 0.16 [a] | nd |
| | +Cd | 89.69 ± 1.05 [e] | 52.94 ± 3.22 [d] | 5.30 ± 0.10 [c,d] | 5.88 ± 0.11 [d] | 0.590 [d] |
| **AS** | −Cd | nd | nd | 6.35 ± 0.12 [b] | 7.25 ± 0.13 [b] | nd |
| | +Cd | 95.59 ± 4.04 [d] | 71.49 ± 1.32 [c,d] | 5.06 ± 0.09 [d] | 5.47 ± 0.10 [e] | 0.747 [b] |
| **Gyp** | −Cd | nd | nd | 5.85 ± 0.11 [c] | 6.92 ± 0.13 [b,c] | nd |
| | +Cd | 109.56 ± 4.62 [c] | 75.62 ± 1.62 [c] | 4.88 ± 0.09 [d] | 5.09 ± 0.09 [f] | 0.694 [c] |
| **MS** | −Cd | nd | nd | 5.27 ± 0.12 [c,d] | 6.37 ± 0.12 [c] | nd |
| | +Cd | 126.84 ± 5.62 [b] | 81.24 ± 1.85 [b] | 4.64 ± 0.08 [d,e] | 4.75 ± 0.09 [f,g] | 0.642 [c,d] |

### 3.4. Effects of Different S Sources on Plant Growth under Cd Stress

Plants subjected to 200Cd showed a decline in fresh biomass by 17.9%, dry biomass by 26.6%, and leaf area by 8.4%, compared to the control (Figure 2). Plants receiving S in different forms showed an increase in all the afore-said growth parameters under no Cd stress, and elemental S showed a maximum increase of 53.6% in fresh biomass, 71.2% in dry biomass, and 56.4% in leaf area, compared to the control. Furthermore, a significant

amelioration of Cd toxicity was seen in plants receiving different S sources. In Cd-exposed plants, the application of elemental S maximally enhanced fresh biomass by 20.6%, dry biomass by 28.7%, and leaf area by 24.5% compared to the control, and so conspicuously restored the damage caused by Cd (Figure 2).

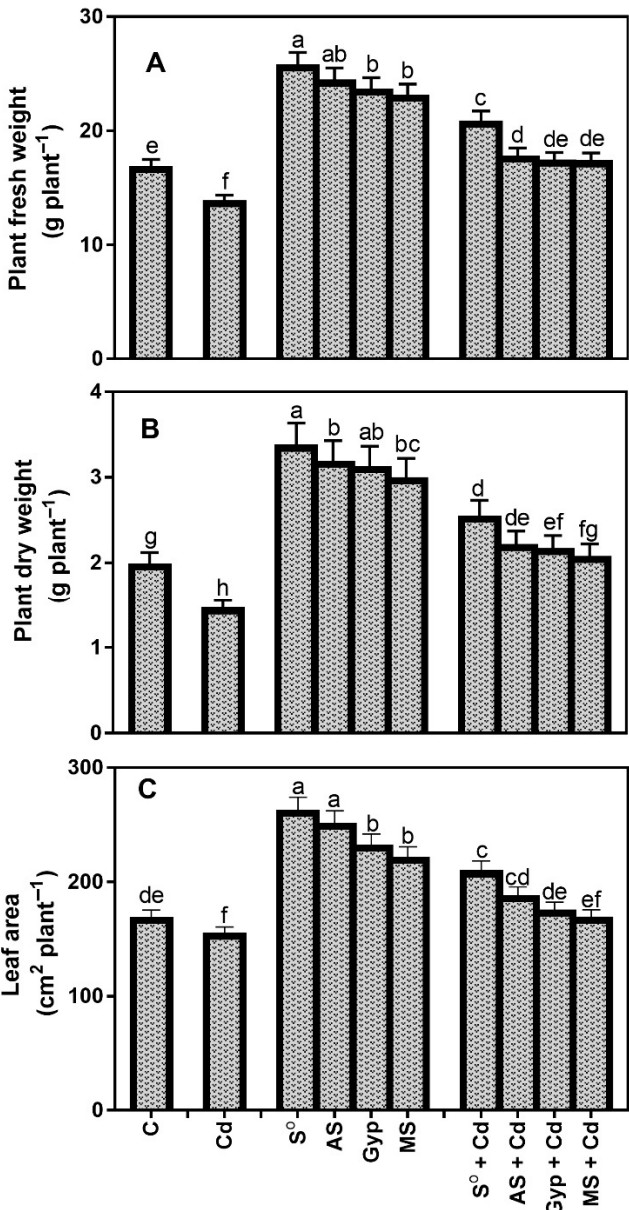

**Figure 2.** (**A**) Plant fresh weight, (**B**) plant dry weight, and (**C**) leaf area in *Brassica juncea* cv. Giriraj supplied with 200 mg S kg$^{-1}$ soil of elemental S (S$^0$), ammonium sulfate (AS), gypsum (Gyp) or magnesium sulfate (MS), grown with or without cadmium (200 mg Cd kg$^{-1}$ soil). Data are presented as means ± SE (*n* = 4). Data are presented as means ± SE (*n* = 4). Data with the same letter are not significantly different by LSD test at *p* < 0.05. C, control; Cd, cadmium.

### 3.5. Effect of Different S Sources in Preventing Adverse Effects of Cd on Photosynthesis

The plants grown in the presence of Cd 200 exhibited reduced net photosynthesis by 22.2%, stomatal conductance by 19.8%, intercellular $CO_2$ concentration by 30.6%, chlorophyll content by 21.2%, maximal PSII efficiency by 18.3%, and Rubisco activity by 27.5%, compared to the control (Figure 3). Although under unstressed conditions, photosynthetic attributes were enhanced by all the four S sources, and elemental S improved theses parameters more prominently than the other S forms. Elemental sulfur supplemental in

plants grown with Cd also resulted in maximal alleviation of Cd-induced photosynthetic inhibition and promoted net photosynthesis, stomatal conductance, intercellular $CO_2$ concentration, and chlorophyll content by 28.9, 64.2, 32.2, and 34.2%, respectively, compared to the control (Figure 3A–D). Similarly, $S^0$ treatment displayed an utmost increase in maximal PSII efficiency and Rubisco activity by 11.1 and 60.5%, respectively, whereas in MS-treated plants, maximal PSII efficiency was reduced by 5.30% and Rubisco activity increased by 20.1%, compared to the control (Figure 3E,F).

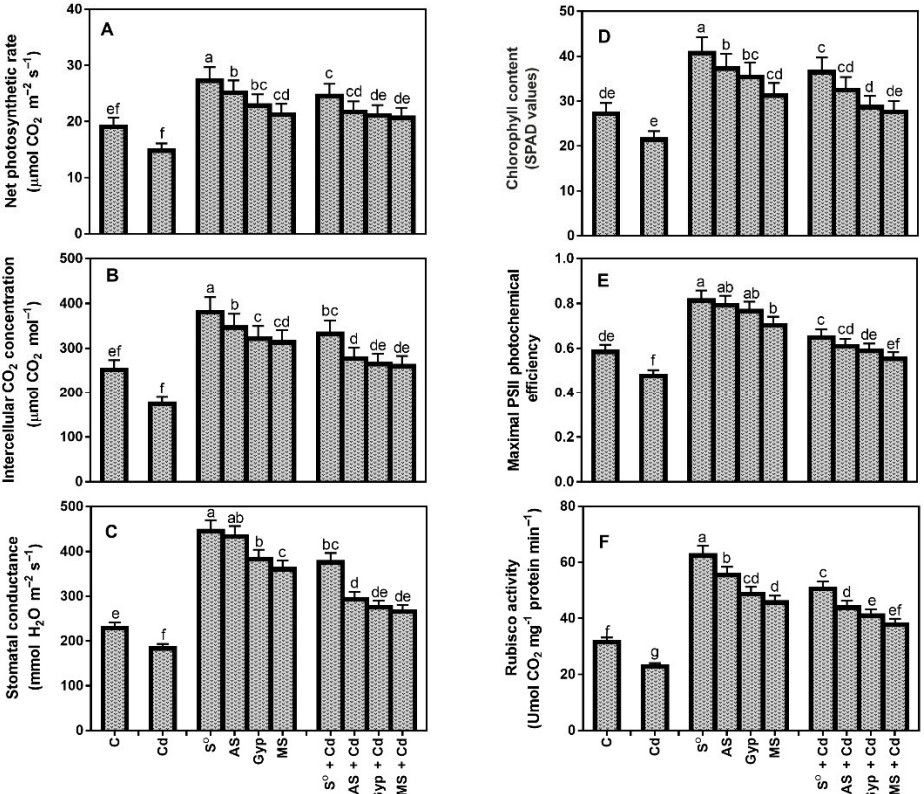

**Figure 3.** (**A**) Net photosynthesis, (**B**) intercellular $CO_2$ concentration, (**C**) chlorophyll content (SPAD values), (**D**) stomatal conductance, (**E**) maximal PSII photochemical efficiency ($F_v/F_m$), and (**F**) Rubisco activity in leaves of *Brassica juncea* cv. Giriraj treated with 200 mg S kg$^{-1}$ soil of elemental S ($S^0$), ammonium sulfate (AS), gypsum (Gyp), or magnesium sulfate (MS), grown with or without cadmium (200 mg Cd kg$^{-1}$ soil). Data are presented as means $\pm$ SE (*n* = 4). Data with the same letter are not significantly different by LSD test at *p* < 0.05. C, control; Cd, cadmium; Rubisco, ribulose-1,5-bisphosphate carboxylase/oxygenase.

### 3.6. Effects of Different S Sources on Oxidative Stress and Antioxidants

Cd presence led to a significant increase in $H_2O_2$ and $O_2^{\bullet-}$ content by 3.5 and 2.8 times, compared to the control (Figure 4). The application of different S sources moderated the production of oxidative markers produced because of Cd toxicity and recovered the oxidative damage, with $S^0$-treated plants showing the most prominent reduction. The accumulation of $O_2^{\bullet-}$ was evidenced by the formation of scattered dark blue formazan in the leaves (Figure 4A–D) and that of $H_2O_2$ by brownish colored formazan in leaves (Figure 4E–H). Fresh leaves from Cd-stressed plants exhibited more pronounced spots when compared to control plants. Furthermore, leaves of plants treated with elemental S without Cd or Cd observed fewer spots as compared to the Cd-stressed plants.

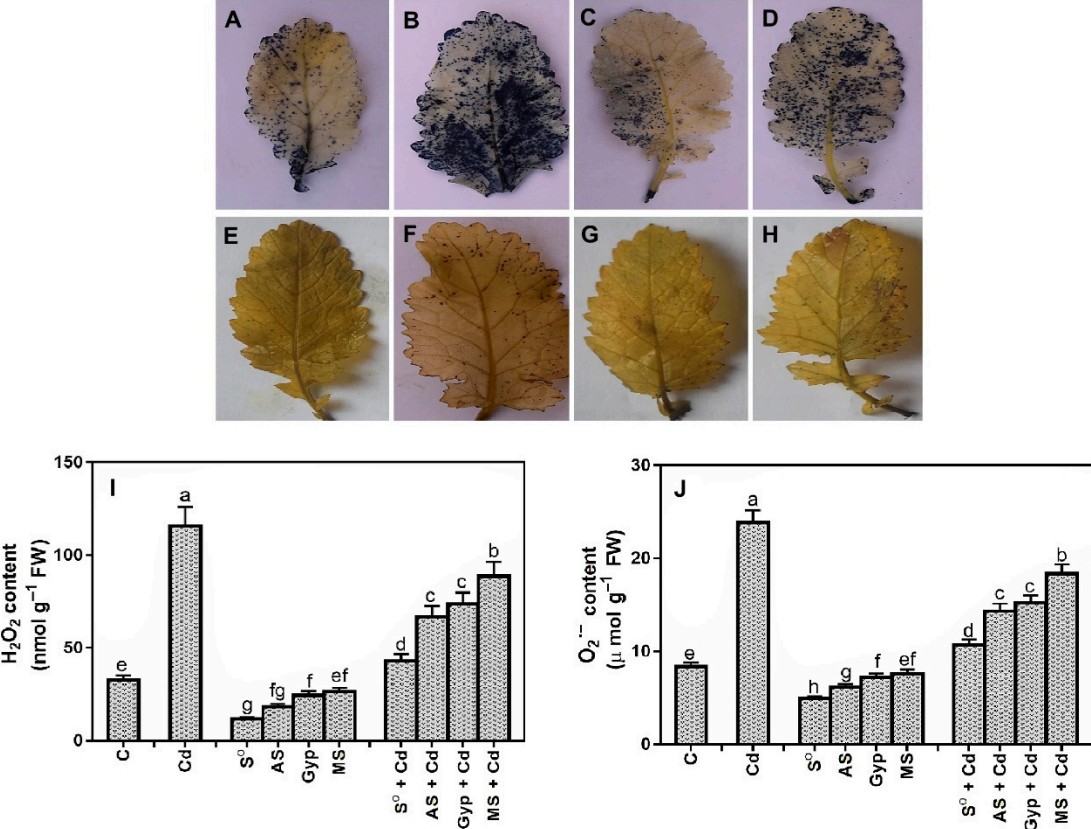

**Figure 4.** Accumulation of superoxide anion ($O_2^{\bullet-}$) and hydrogen peroxide ($H_2O_2$) in leaves of *Brassica juncea* cv Giriraj stained with (**A–D**) NBT and (**E–H**) DAB, respectively, under (**A,E**) control 0 mg Cd kg$^{-1}$ soil + 0 mg S kg$^{-1}$ soil, (**B,F**) 200 mg Cd kg$^{-1}$ soil, (**C,G**) 200 mg S kg$^{-1}$ soil, and (**D,H**) 200 mg Cd kg$^{-1}$ soil + 200 mg S kg$^{-1}$ soil. (**I**) $H_2O_2$ content and (**J**) $O_2^{\bullet-}$ content in leaves of *Brassica juncea* cv Giriraj treated with 200 mg S kg$^{-1}$ soil of elemental S ($S^0$), ammonium sulfate (AS), gypsum (Gyp) or magnesium sulfate (MS), grown with or without cadmium (200 mg Cd kg$^{-1}$ soil). Data are presented as means $\pm$ SE (*n* = 4). Data with the same letter are not significantly different by LSD test at *p* < 0.05. C, control; Cd, cadmium; DAB, 3,3′-diamiobenzidine; FW, fresh weight, NBT, nitro blue tetrazolium.

Cd stress significantly raised the electrolyte leakage and content of thiobarbituric acid reactive substances (TBARS) by 2.2 and 2.6 times, compared to control plants (Figure 5A,B). Nevertheless, distinct S sources supplementation proved efficacious in curbing Cd-induced increase in electrolyte leakage and lipid peroxidation and toned down both stress biomarkers efficiently. Moreover, Cd-exposed plants with elemental S exhibited a decrease by 19.5% in electrolyte leakage and by 21.7% in TBARS content, with respect to control plants. The reduction in electrolyte leakage and TBARS content was lower in MS-treated plants, which displayed a considerable increase among all the S sources under Cd stress, compared to the control (Figure 5A,B).

The oxidative stress test results suggested that the exogenous treatment with S sources helped plants to tolerate Cd stress by deescalating Cd-induced oxidative injury caused by ROS accumulation (Figure 5C–E). Indeed, plants exposed to Cd showed an increase in the activities of CAT, SOD, and APX enzymes by 91.5, 39.1, and 52.8%, compared to the control, and this increase was significantly higher than in unstressed plants supplied with different S sources, showing the response of plants' inherent defense capability to counter oxidative stress (Figure 5C–E). Plants treated with various S forms exhibited an increase in activity of all the three antioxidant enzymes under Cd presence. The activity of CAT, SOD, and APX was highest with $S^0$ (3.5, 3.8 and 6.3 times, respectively), with respect to the control (Figure 5C–E).

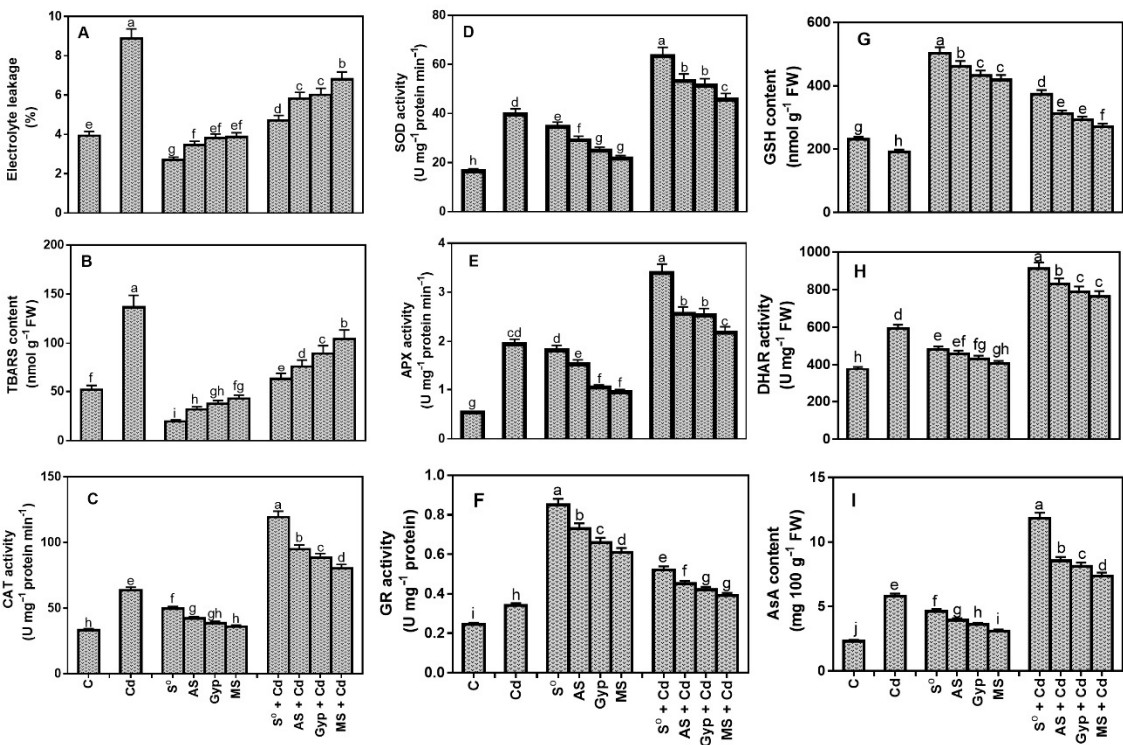

**Figure 5.** Foliar (**A**) Electrolyte leakage, (**B**) TBARS content, (**C**) CAT activity, (**D**) SOD activity, (**E**) APX activity, (**F**) GR activity, (**G**) GSH content, (**H**) DHAR activity, and (**I**) AsA content (**I**) in *Brassica juncea* cv. Giriraj treated with 200 mg S kg$^{-1}$ soil of elemental S (S$^0$), ammonium sulfate (AS), gypsum (Gyp) or magnesium sulfate (MS), grown with or without cadmium (200 mg Cd kg$^{-1}$ soil). Data are presented as means $\pm$ SE (*n* = 4). Data with the same letter are not significantly different by LSD test at *p* < 0.05. APX, ascorbate peroxidase; AsA, ascorbate; C, control; CAT, catalase; Cd, cadmium; DHAR, dehydroascorbate reductase; FW, fresh weight; GR, glutathione reductase; GSH, reduced glutathione; SOD, superoxide dismutase; TBARS, thiobarbituric acid reactive substances.

In order to assess the role of the AsA-GSH cycle in Cd stress tolerance, assessment of activity of GR and DHAR and content of AsA and GSH were evaluated. The Cd-treated plants exhibited an increase in GR activity by 39.3%, while GSH content was reduced by 16.8%, compared to the control (Figure 5F,G). Plants grown with different sources of S increased activity of GR and GSH content in normal as well as in stressed conditions compared to the control. Plants receiving S$^0$ with Cd maximally increased the activity of GR and content of GSH by 2.1 and 1.6 times, respectively, compared to control plants. Similarly, the activity of DHAR and AsA contents under the influence of distinct S sources increased, and the maximal increase was recorded in S$^0$-treated plants, in both stressed as well as unstressed conditions (Figure 5H,I).

### 3.7. Effect of Different Sources of S on Variations in S Assimilation under Cd Stress

Plants grown with Cd exhibited increases in the activity of ATP-S and OASTL by 29.4 and 37.5%, respectively, compared to the control (Figure 6A,B). The activity of both the S-assimilation enzymes was significantly elevated by all the S sources, in unstressed as well as in stressed plants. Furthermore, plants treated with Cd plus distinct S forms showed a significant and marked increase in the activities of ATP-S and OASTL by 3.2 and 3.8 times using S$^0$, 2.8 and 3.4 times with AS, 2.5 and 3.2 times with Gyp, and 2.3 and 2.7 times by supplying MS, respectively, compared to the control (Figure 6A,B). The content of Cys and Met increased with all the S sources, with or without Cd (Figure 6C,D). Plants grown under Cd showed increased Cys content by 53.2%, whereas Met content was reduced by 24.9%, compared to control plants (Figure 6C,D). Elemental S maximally improved the content of Cys and Met by 2.8 times in unstress plants, with respect to the control. Further, S sources

reversed the Cd-induced reduction in Cys and Met content and improved it maximally by 4.3 and 1.9 times, respectively, using $S^0$.

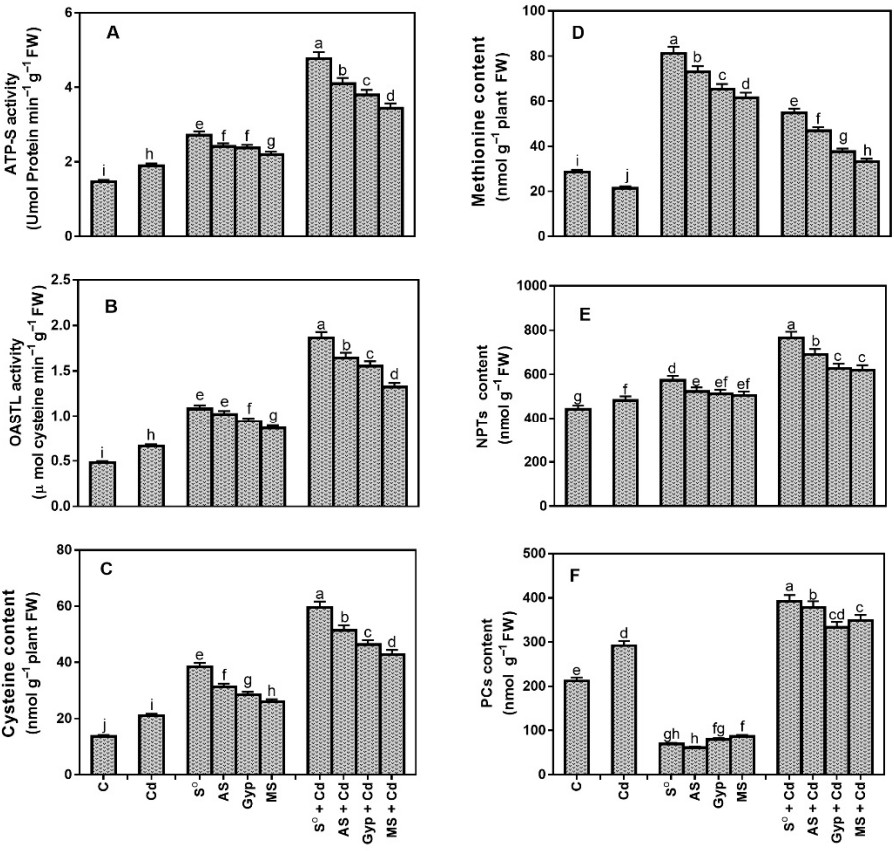

**Figure 6.** Foliar (**A**) ATP-S activity, (**B**) OASTL activity, (**C**) Cys content, (**D**) Met content, (**E**) NPT content, and (**F**) PCs content in *Brassica juncea* cv. Giriraj treated with 200 mg S $kg^{-1}$ soil of elemental S ($S^0$), ammonium sulfate (AS), gypsum (Gyp), and magnesium sulfate (MS), grown with or without cadmium (200 mg Cd $kg^{-1}$ soil). Data are presented as means ± SE (*n* = 4). Data with the same letter are not significantly different by LSD test at *p* < 0.05. ATP-S, ATP sulfurylase; C, control; Cd, cadmium; FW, fresh weight; NPTs, non-protein thiols; OASTL, O-acetylserine (thiol) lyase; PCs, phytochelatins.

NPT and PCs content was analyzed in leaves of both stressed and unstressed plants. Both NPT and PCs content in Cd-stressed plants was significantly higher than in control plants (9.2 and 83.4%, respectively) (Figure 6EF). Plants treated with different forms of S under Cd stress showed a significant and conspicuous rise in the content of both NPT and PCs. NPT content, which was maximal in the plants treated with $S^0$ + Cd (1.7 times), was followed by AS-treated plants (1.5 times), whereas the plants treated with Gyp/MS + Cd displayed no significant difference over control plants (Figure 6E). Similarly, PC content also showed a significant and remarkable increase by 1.8 times following $S^0$ + Cd treatment, while Gyp + Cd ranked lowest (1.5 times) in increasing PC content among the four S sources used, compared to control plants (Figure 6F).

### 3.8. Influence of Different S Sources on Cell Viability and Stomatal Studies under Cd Stress

Propidium iodide (PI) is a staining dye that penetrates damaged cell membranes and stains nucleic acids, which become visible inside the dead cell as red fluorescent spots. Root cells of Cd-stressed plants were less viable and showed more stain. However, Cd-induced cell death was reversed by S application, and $S^0$-treated plants exhibited a similar response to control plants (Figure 7A–D).

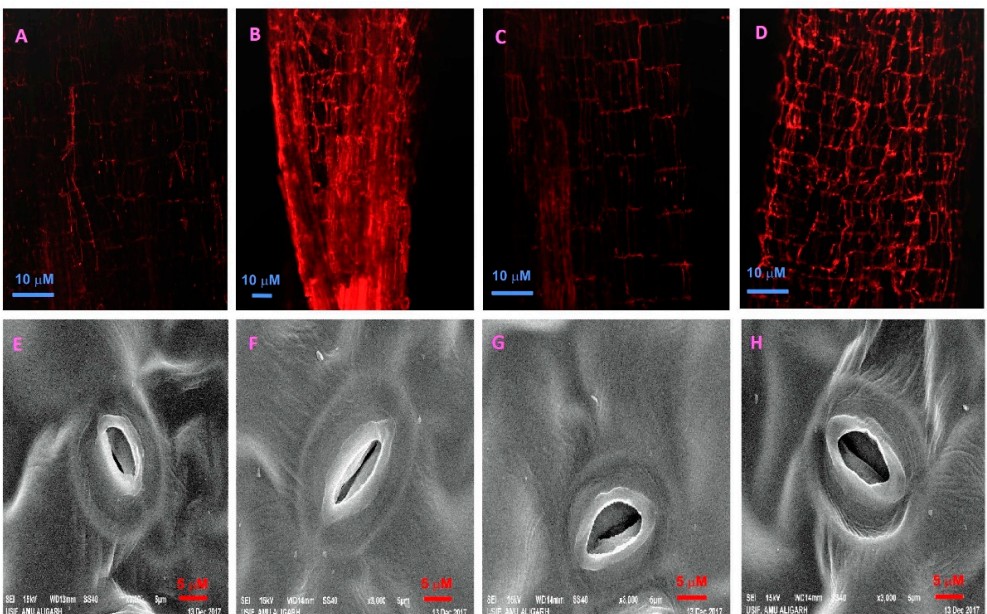

**Figure 7.** (**A–D**) Cell viability test of roots cells and (**E–H**) leaf stomatal response of *Brassica juncea* cv. Girira under (**A,E**) control 0 mg Cd kg$^{-1}$ soil + 0 mg S kg$^{-1}$ soil, (**B,F**) 200 mg Cd kg$^{-1}$ soil, (**C,G**) 200 mg S kg$^{-1}$ soil, and (**D,H**) 200 mg Cd kg$^{-1}$ soil + 200 mg S kg$^{-1}$ soil. Bars (**A–D**) = 10 μM; bars (**E–H**) = 5 μM.

Electron microscopy was used to examine the stomatal behavior in response to Cd stress and S supply. Stomatal analysis depicted a noteworthy change in stomatal aperture in Cd-stressed plants. Stomata were normal in the leaf samples of control, S$^0$ and Cd + S$^0$-treated plants. However, Cd-treated plants showed semi-closed stomata with distorted guard cells (Figure 7E–H).

## 4. Discussion

The present study aimed to scrutinize the efficacy of different sources and levels of S in influencing growth and photosynthesis and antioxidant metabolism, and to investigate the potential of the S sources in the alleviation of Cd-induced stress in *B. juncea*. Dose-dependent S requirement revealed that 200 mg kg$^{-1}$ S (200S) was more effective for achieving maximum growth and photosynthesis than 100S. The S sources used showed a differential effect in mitigating Cd-induced toxicity in plants. In general, S$^0$ was the most effective among all the four S sources used, followed by AS, then Gyp and lastly MS.

### 4.1. S-Assimilation Plays a Central Role in Enhancing Defense against Cd Stress

Sulfate uptake followed by S-assimilation is presumed to be a critical determinant for plant's survival to a wide range of environmental cues since S-containing compounds confer protection to both biotic and abiotic stresses [9,10,12–14]. In the present study, the use of all four different S sources increased S-assimilation enzymes, ATP-S and OASTL, and S-containing amino acid content (Cys and Met), which were able to detoxify Cd (Table 4 and Figure 6). Our results agree with the studies of Liang et al. [15] and Lou et al. [36] in *Brassica chinensis*, Khan et al. [6] in wheat, Per et al. [40,41], Asgher et al. [42], Masood et al. [21] in mustard, and Flávio et al. [43] in maize, where S assimilation was significantly involved in lowering Cd-induced toxicity by increasing contents of enzymatic and non-enzymatic antioxidants and S-containing metabolites. It is also known that S-assimilation is upregulated in response to various biotic and abiotic factors [44]. Elemental S was also observed to be effective in enhancing the S-assimilation in some previous studies focused on salt-stressed mustard [45,46] and mung bean [47]. Elemental sulfur is a slow S releasing fertilizer with high S content and, when compared with other S forms, it increases the soluble SO$_4^{2-}$ content in soil due to microbial oxidation [48]. Among essential elements, mobility of S is intermediate, as it is less mobile compared to N, P, and K, but mobile compared to

Ca and most micronutrients. So, a continuous supply of S is required for emergence to crop maturity. Moreover, $S^0$ has a long time of residence in soil and can supply till later stages of growth, while other forms of S release $SO_4^{2-}$, which is mostly available at the earlier stages of growth and gets easily depleted [48]. S assimilation in plants is the principal biosynthetic pathway leading to amino acids, which are required for the synthesis of various metabolites, such as GSH, NPT, and PCs, which contribute in mechanisms of biochemical adaptation to metal stress [3]. These S-containing amino-acid pool also participate in enhancing the biosynthesis of antioxidants [17,18]. Thus, $S^0$-assisted rises in antioxidants, along with S-metabolites, are the key players in conferring Cd tolerance to *B. juncea* plants. The enhanced S assimilation could be the reason for mitigating Cd toxicity, as it contributed in the production of GSH (Figure 5G), Cys, and Met (Figure 6C,D), and also increased the activity of ATP-S and OASTL (Figure 6A,B). Moreover, ATP-S overexpression was also reported to improve the phytoextraction capability of *B. juncea* plants for Cd [49]. A study by Xiang and Oliver [50] in Arabidopsis showed that, in response to Cd stress, plants responded by overexpressing genes involved in S-assimilation and in the synthesis of GSH and PCs. Hence, our results showed that supplemental S nutrition upregulated S-assimilatory pathway and increased the status of antioxidant metabolites, such as GSH (Figure 5G) and of PCs (Figure 6F), which are oligomers of GSH and act as heavy metal chelators and/or detoxifiers, and play a prominent role in the mitigation of Cd-induced stress.

### 4.2. Cd Accumulation, Translocation, and Role of Sulfur

Cadmium concentration was highest in roots and lower in leaves of *B. juncea*, as roots are the prime organ of Cd accumulation (Table 3). This is in accordance with previous findings by Zhang et al. [51], Per et al. [41] and Yamaguchi et al. [52]. Among the four S sources used, $S^0$ caused the lowest Cd levels in roots and leaves, as confirmed by the low values of translocation factor (TF) (Table 4). This decrease in Cd accumulation after the application of $S^0$ could be due to immobilization of Cd in soil, which reduces its availability to mustard plants [17]. The addition of $S^0$ to alkaline soils, compared to other S sources, reduces soil pH through microbial oxidation [53,54]. This increases the solubility of micronutrients, which becomes readily available to plants. On the other side, ammonium sulfate, gypsum, and magnesium sulfate have a little or no considerable effects on soil pH. Moreover, in alkaline soils, free $SO_4^{2-}$ of ammonium sulfate, gypsum, and magnesium sulfate can form complexes with Cd, forming $SO_4^{2-}$-Cd complexes in Cd-polluted soils [55]. Therefore, $SO_4^{2-}$ in the form of ammonium sulfate, gypsum, and magnesium sulfate become less available for plant uptake. Moreover, $S^0$ ensures enough availability of $SO_4^{2-}$ by microbial action to be used by plants for longer times. On the contrary, Zhang et al. [19] found that gypsum was more effective in reducing Cd accumulation than $S^0$ in rice plants. However, there are many mechanisms by which a plant minimizes Cd toxicity, including immobilization (binding of Cd to cell wall), compartmentalization (vacuolar sequestration), prevention (decreasing Cd entry into roots), and chelation (Cd-PCs mediated) [12]. In the present study, S supply enhanced tolerance to *B. juncea* plants against Cd by accumulating more Cd in roots than shoots (Table 4), reducing Cd translocation by the involvement of heavy metal chelators, like NPTs and PCs (Figure 7E,F) (Table 3). These observations coincide with the works of Yamaguchi et al. [52,56] and Liang et al. [15], where the authors reported the S-mediated induction of limited translocation Cd in the tested plants. Besides, there are also several reports that have shown that S supply induces sulfate uptake transporters and the level of their transcripts, which resulted in increased S content in Cd-exposed plants [52,57,58]. Besides, it has been reported that $S^0$-assisted microbial oxidation increases the availability of other minerals, such as Fe, N, Cu, Ca, and Mn in soil [59]. Therefore, $S^0$ supplement prevents the deficiency of mineral elements, particularly in Cd-stressed conditions. S mediated uptake of sulfate along with other nutrients could also be another major strategy for reducing Cd uptake from soil and lower Cd toxicity.

### 4.3. Sulfur Increases Plant Growth by Mitigating Cd-Induced Toxicity

In the present study, Cd-induced stress resulted in the reduction of the major growth parameters like plant fresh biomass, plant dry biomass, and leaf area (Figure 2). S supplementation modulated Cd-induced toxicity in mustard and improved growth biomarkers, with $S^0$ being the most effective among the four S sources used (Figure 2). The increase in plant biomass specifically with $S^0$ could be attributed to the reduction in Cd-induced phytotoxicity [60]. Notably, soil pH is the most important parameter controlling mobility of cationic heavy metals in soils [61]. Lowering the soil pH increases the solubilization of Cd into the soil. Microbial oxidation of $S^0$ introduced into the soil leads to the production of $H_2SO_4$. This process precipitates divalent cationic metals such as Cd [62,63]. This could also be one of the reasons for higher plant growth with $S^0$ supplementation. In contrast, Singh et al. [64] reported that rice plants supplied with bentonite S exhibited greater biomass and outperformed other S sources, namely $S^0$, ordinary super phosphate and gypsum. However, in our study, the enhancement in growth with S under Cd stress is parallel to previous studies reported for *B. campestris* [17], *Oryza sativa* [6], *Fagopyrum tararicum* [12], and *B. juncea* [41]. Sulfur can amend the growth through inhibiting oxidative stress and restraining Cd transport from root to shoot [16]. Besides this, Asgher et al. [42] showed that plants exposed to Cd and exogenous S possessed more plant dry biomass, compared to Cd that only stressed plants. Treatment with ammonium phosphate and S fertilizers were used for reduction in Cd translocation, which advanced shoot growth in rice plants [9]. Moreover, S treatment enhanced total plant fresh weight, plant height, and root length in *B. chinensis*, thereby decreasing the inhibitory effects of Cd toxicity on plant growth [36]. These findings suggest that diminished plant growth is an indicator and a direct measure of Cd toxicity and that S has a positive role in influencing plant growth under Cd stress.

### 4.4. Sulfur Prevents Negative Effects of Cd on Photosynthesis

Cd may exert its inhibitory effects on photosynthesis directly by interacting with its enzymes [5], or indirectly by obstructing the synthesis of photosynthetic pigments or by causing their degradation [8]. In our study, Cd-treated plants showed a considerable reduction in chlorophyll content, leaf gas exchange parameters, maximal PSII efficiency, and Rubisco activity (Table 3 and Figure 3). The reversal of damage on photosynthetic characteristics and photo-inhibition caused by Cd was modulated by exogenous application with various S sources (Table 3 and Figure 3). The results show that all S sources increased various photosynthetic attributes, which were earlier reduced because of Cd-induced phytotoxicity. Even in this case, $S^0$ showed the highest beneficial effects. Elemental S has been used in many studies in improving photosynthesis under various types of abiotic stresses [19,45]. The reason for increased photosynthesis under Cd stress with elemental S involves S-mediated allocation of N to Rubisco protein and through increase in stomatal conductance [41]. Sulfur starvation aggravates chlorophyll content, maximal PSII efficiency, Rubisco activity, and causes alterations in photosynthetic electron carriers, loss of grana, while S deficiency under Cd stress resulted in severe disintegration of thylakoids, suggesting the importance of S nutrition in photosynthesis [65]. Nazar et al. [66] reviewed the involvement of excess S in improving quantum yield, leaf gas exchange parameters, chlorophyll content, and Rubisco activity due to the enhanced synthesis of GSH, which provides a shielding effect to photosynthetic apparatus against salt stress. This finding suggests that the application of S can enhance photosynthesis by reversal of damage on common attributes associated with photosynthetic efficiency. This beneficial effect of S was visible in the response of guard cells to S, which allowed stomatal opening also in the presence of Cd (Figure 7H).

### 4.5. Sulfur Is Involved in the Reversal of Cd-Induced Oxidative Burst

Cd exerts its harmful effects by causing oxidative injury to plants triggered by ROS accumulation [6,41,42,67]. Our results show that Cd exposure in surplus quantity severely

increased ROS formation ($O_2^{\bullet-}$, $H_2O_2$), electrolyte leakage (Figure 5A) and lipid peroxidation (Figure 5B) in leaves of *B. juncea* plants (Figure 4). A significant reduction in oxidative biomarkers was observed in plants treated with different S forms + Cd, suggesting that S supply protected plant cells from oxidative injury. Antioxidant enzymes are used by plants to scavenge excess ROS production under harsh environmental conditions, thus preventing cells from oxidative damage. The current study showed that Cd supply caused a small increase in antioxidant enzyme activity (CAT, SOD, APX), however, the application of S boosted this upsurge in antioxidant defense, which is particularly clear in plants treated with $S^0$ + Cd (Figure 5C–E). Earlier reports have also shown that S treatment triggered the stimulation of antioxidants (SOD, CAT, APX), which played an active role in defending various environment cues like Cd toxicity [6,18,40,41], salt stress [45,46], and Cr (VI) excess [68].

The AsA-GSH cycle, also known as Foyer-Halliwell-Asada cycle, is a major pathway in maintaining a proper balance between generation and quenching of excessive ROS and regulating cellular redox status inside the cell in plants facing various environmental challenges [17,68]. In our study, GR and DHAR activities increased in response to Cd stress in plants supplied with various S sources (Figure 5F,H), and this was consistent with the increase GSH and AsA content (Figure 5G,I). The maximal beneficial effect to plants was obtained with the application of $S^0$, where it completely outperformed the other three sources (ammonium sulfate, gypsum and magnesium sulfate), both in the presence or absence of Cd. A plethora of literature scrutinizes the efficiency of AsA-GSH cycle in alleviating Cd-induced oxidative stress in response to S supply [12,15,17]. GSH acts as the first line of defense in response to Cd-induced toxicity before the synthesis and arrival of PCs [69]. The reversal of Cd-induced oxidative burst is clear by the visibly increased cell viability test of roots cells in plants subjected to S treatment and Cd exposure, compared to the roots of only Cd-stressed plants (Figure 7D).

Our study also revealed that Cd exposure reduced GSH levels in only Cd-treated plants, compared to control plants (Figure 5G), and this agrees with the study of Liang et al. [15]. This depletion in the GSH pool can be correlated to demand in the synthesis of PCs, which is an integral component of NPTs and can be reversed by application with S [36], which is in harmony with our results (Figure 6E,F). Therefore, we can affirm that S application relieved Cd-induced oxidative stress by upregulating plant's antioxidant machinery, by improving the AsA-GSH pathway and by inducing the biosynthesis of heavy metal chelators, like NPT and PCs, which detoxified Cd and hence lowered oxidative stress (Figure 8).

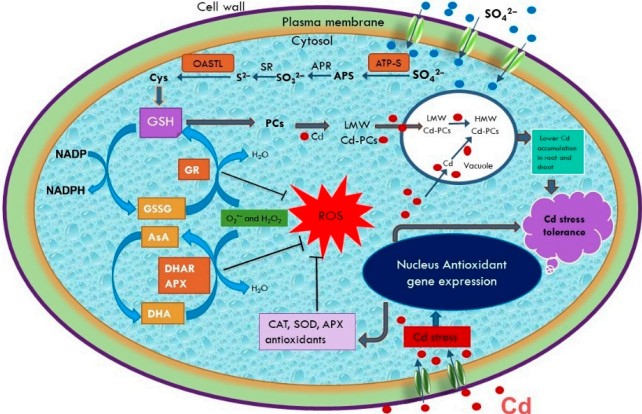

**Figure 8.** Simplified representation of the major mechanisms and signal transduction pathways underlying sulfur-mediated alleviation of cadmium stress in plants. HMW, high molecular weight; LMW, low molecular weight. The other acronyms in the figure are explained in the text.

## 5. Conclusions

In this study, 200S treatment was more effective than 100S in counteracting Cd stress in *B. juncea*. The S supply reduced Cd uptake and Cd accumulation in root and leaves, with plants showing lower oxidative stress symptoms. Sulfur given to plants significantly increased enzymatic and non-enzymatic antioxidant defense, and the levels of PCs and NPTs, which are both useful for the detoxification and sequestration of Cd, respectively. The results also showed the role of the different S sources (elemental S, ammonium sulfate, gypsum, and magnesium sulfate) and their response towards Cd, with $S^0$ showing the more significant effects among all S sources in detoxifying Cd-induced phytotoxicity. Elemental S as a source of S was used in many studies to scrutinize its efficiency in Cd-exposed plants [41,55,60] and it is recommended over other organic S-based fertilizers since it (i) has a longer time of residence in soil, (ii) can activate soil microbiome, (iii) can enhance the productivity and the nutritional quality of crops, and (iv) can transform into reactive sulfur species and volatile sulfur species that can, in turn, enhance plant tolerance to environmental stresses [48].

The problem of Cd contamination of agricultural soil is increasing continuously because of its addition through phosphatic fertilizers, contaminated irrigation or other sources. It is so urgent and essential to find strategies and mechanisms that can lower the effects of Cd toxicity in crops. The application of sulfur could be promoted in soils contaminated with subtoxic or toxic Cd levels to improve the growth and development of cultivated plants. The mechanisms on which this higher defense and tolerance of plants against Cd is based may be explored further through biotechnological and genetic tools.

**Author Contributions:** Conceptualization: A.M. (Asim Masood) and N.A.K.; Investigation and data curation: I.R.M., B.A.R., A.M. (Asim Masood), Z.S.; Cytological and histological analysis: A.S.; Biochemical analysis: I.R.M., B.A.R., A.M. (Asim Masood); Physiological analysis: A.M. (Arif Majid), N.A.A.; Original draft preparation: I.R.M. and B.A.R.; Editing and content improvement: A.S., I.D., A.M. (Asim Masood), N.A.K. and N.A.A. All authors have read and agreed to the published version of the manuscript.

**Funding:** This research was funded by Department of Science and Technology SERB (Project code: SB/YS/LS-108/2014), New Delhi India awarded to A.M. for advancing the laboratory facilities.

**Institutional Review Board Statement:** Not applicable.

**Informed Consent Statement:** Not applicable.

**Data Availability Statement:** Raw data of this article are available upon request to corresponding authors.

**Acknowledgments:** The Authors are also grateful to the University Sophisticated Instruments Facility (USIF) of the Aligarh Muslim University and to the University of Basilicata (UNIBAS-DiCEM) for providing the necessary instrument facilities.

**Conflicts of Interest:** The authors declare no conflict of interest.

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
