# Peer review of "Soil Sulfur Sources Differentially Enhance Cadmium Tolerance in Indian Mustard (Brassica juncea L.)"

_soilsystems, doi:10.3390/soilsystems5020029_

Round 1

Reviewer 1 Report

I would like to congratulate the authors for their work that confirms the role of sulfur in alleviating cadmium phytotoxicity. I really appreciated the experiments and analysis carried out in this study.
Please find a list of minor revisions which I hope the authors might find useful to improve the overall quality of their work.

Chapter 2.1: I suggest dividing the description of the three experiments in order to improve the understanding of the text.

Line 107: "The cultivar Giriraj emerged as a Cd-tolerant" is a result.

Line 133: "Cd content was determined by atomic absorption spectrophotometry" was just reported at line 130.

Chapter 3.1: Please carefully review the text. The highest Cd accumulation was recorded in Pusa tarak and not Pusa Agrani. Moreover, sometimes the mustard cultivar is called Pusa Agrani, and other times Pusa Agrini. I suggest adding a column with Tolerance Index (TI) in Table 2 and eliminate Figure 1.

Line 287: Are 48.9, 37.7, 75.2, 70.9, 35.2 40.1, and 86.3 percentages?

Lines 313-315: The sentence is not clear, please reformulate it.

Line 355-356: This sentence appears redundant with the results reported at lines 357-359.

Line 391: Please add the treatment of pictures E, F, G, H. 

Among the papers included in the references, 17 out of 63 are attributable to authors, I believe that they are excessive and not fundamental for understanding this study. 

Author Response

I would like to congratulate the authors for their work that confirms the role of sulfur in alleviating cadmium phytotoxicity. I really appreciated the experiments and analysis carried out in this study.

Thank you for your appreciation and encouragement. We followed your suggestions and corrections and now we hope that this revised version has been improved.

Please find a list of minor revisions which I hope the authors might find useful to improve the overall quality of their work.

Thank you, here below a point-by-point list of our changes.

Chapter 2.1: I suggest dividing the description of the three experiments in order to improve the understanding of the text.

We have divided the paragraph 2.1 in sections where the experiments were clarified.

Line 107: "The cultivar Giriraj emerged as a Cd-tolerant" is a result.

Thank you. We deleted this sentence from this section.

Line 133: "Cd content was determined by atomic absorption spectrophotometry" was just reported at line 130.

Thank you. We changed this sentence.

Chapter 3.1: Please carefully review the text. The highest Cd accumulation was recorded in Pusa tarak and not Pusa Agrani. Moreover, sometimes the mustard cultivar is called Pusa Agrani, and other times Pusa Agrini. I suggest adding a column with Tolerance Index (TI) in Table 2 and eliminate Figure 1.

We added a column with TI values and deleted Figure 1.

Line 287: Are 48.9, 37.7, 75.2, 70.9, 35.2 40.1, and 86.3 percentages?

Yes, they are percentages.

Lines 313-315: The sentence is not clear, please reformulate it.

Yes, we agree. We shortened and clarified it.

Line 355-356: This sentence appears redundant with the results reported at lines 357-359.

We agree and condensed the two sentences.

Line 391: Please add the treatment of pictures E, F, G, H.

We added them in the figure caption.

Among the papers included in the references, 17 out of 63 are attributable to authors, I believe that they are excessive and not fundamental for understanding this study.

Right. We reduced the number of references.

Kind regards

Reviewer 2 Report

The present study describes that sulfur fertilization enhance cadmium tolerance in Indian mustard by increasing antioxidative responses and compounds for chelation of Cd. Their experiments are so detail with measurements of various physiological responses in plants such as photosynthesis, stomata, enzymes and metabolites in S-assimilation.  

However, enhancement of Cd tolerance in plants by S application is not new fact, as this has been observed in various previous studies. Their new points are different responses by different S sources; thus, the authors should discuss more why elemental S confer more Cd tolerance to plants than other S sources. In discussion, they just compared their results with previous results. They have to discuss more about the mechanisms of elemental S for higher Cd tolerance compared to other sources. Enhanced antioxidative responses are not the mechanisms, but they are observed responses. Authors have to discuss why elemental S confer higher antioxidative responses and higher S containing metabolites compared to other S sources.

For example, elemental S is known to decrease soil pH. Thus the authors can discuss about pH changes in soils by different S source application. If possible, show the soil pH after plant sampling. Results of soil analysis in Table 1 show that the soils they used are weak alkaline before plant cultivation, thus it is possible that application of elemental S at 15 days before sowing may have decreased soil pH. Lowered soil pH should have increased solubility of micronutrient, is this related to Cd tolerance?

I am also wondering if elemental S also can induce higher Cd tolerance for acid soils. The authors should discuss if elemental S is also effective on soils with different pH than that in the present study.

In conclusion (line 624) they describe that elemental S has a longer time of residence in soil. Is it related to higher Cd tolerance? I am wondering if their pots have holes on the bottom or not. If the pots have holes, sulfate ions form of ammonium sulfate, gypsum and magnesium sulfate would have been easily drained off while S in elemental sulfur remained longer. If there is no hole at the bottom of the pots, sulfate should not be drained off. They should discuss this possibility by describing about the presence of holes in the pots.

If there are other possible mechanisms for the higher Cd tolerance by elemental S, they should describe in discussion.

Followings are minor points.

Line 103: Describe the lengths of day and night.

Line 108: (100S, lowS) (200S high S)

Actually, 100 mg S kg-1 soil is not low amount as S fertilizer. So, it is strange to write as “low”. Just 100S and 200S are enough.

Lint 112: So

“0“ should be superscript.

Table 1

Experiment 1, 2, and 3 are only described in this table. Show also in the texts in Materials and Methods which is Experiment 1, 2 and 3.

Figure 1

Is this pie chart really tolerance index? There is not “means + SE” in the pie chart as in the legend. And “Giriraj had the highest tolerance index (0.789) (Line 248)” is not shown in this figure.

Table 2

The unit for Cd level should not be “mg S kg -1 soil”. It should be “mg Cd kg -1 soil”.

Table 2 and Table 4

Delete “g” or “e” from “nd” because it is impossible to perform statistical analysis with “nd”.

Figure 8 legend Line 477-478  Use “,” in place of “-”.

“under (A-E) control 0 mg Cd kg -1 soil + 0 mg S kg -1 soil, (B-F) 200 mg Cd kg -1  soil, (C-G) 200 mg S kg-1 soil, and (D-H) 200 mg Cd kg -1 soil + 200 mg S kg -1 soil.

should be

under (A, E) control 0 mg Cd kg-1 soil + 0 mg S kg-1  soil, (B, F) 200 mg Cd kg-1  soil, (C, G) 200 mg S kg-1  soil, and (D, H) 200 mg Cd kg-1  soil + 200 mg S kg-1  soil.

Line 523 more Cd in roots than shoots (Table 2)

Table 2 should be Table 4.

Author Response

The present study describes that sulfur fertilization enhance cadmium tolerance in Indian mustard by increasing antioxidative responses and compounds for chelation of Cd. Their experiments are so detail with measurements of various physiological responses in plants such as photosynthesis, stomata, enzymes and metabolites in S-assimilation.

Thank you for your revision. We followed your suggestions and corrections and now we hope that this revised version has been improved.

However, enhancement of Cd tolerance in plants by S application is not new fact, as this has been observed in various previous studies. Their new points are different responses by different S sources; thus, the authors should discuss more why elemental S confer more Cd tolerance to plants than other S sources. In discussion, they just compared their results with previous results. They have to discuss more about the mechanisms of elemental S for higher Cd tolerance compared to other sources. Enhanced antioxidative responses are not the mechanisms, but they are observed responses. Authors have to discuss why elemental S confer higher antioxidative responses and higher S containing metabolites compared to other S sources.

We agree and changed the discussion accordingly.

So is a slow S releasing fertilizer with high S content and when compared with other S forms, So treatment increases the soluble SO42- content in soil due to microbial oxidation [57]. S is immobile in the plants, thus a continuous supply of S is required for emergence to crop maturity. Since, So has a long time of residence in soil and can supply till later stages of growth while the SO42- available from the other S sources is mostly available at the earlier stages of plant and gets depleted. S assimilation in plants is the principal biosynthetic pathway leading to amino acids, which are required for synthesis of various metabolites such as GSH, NPT and PCs that contribute in mechanisms of biochemical adaptation to heavy metal stress [3]. These S-containing amino-acid pool also participate in enhancing the biosynthesis of antioxidants [17, 18]. Thus So assisted rise in antioxidants along with S-metabolites are the key players in conferring Cd tolerance to B.juncea plants.

For example, elemental S is known to decrease soil pH. Thus the authors can discuss about pH changes in soils by different S source application. If possible, show the soil pH after plant sampling. Results of soil analysis in Table 1 show that the soils they used are weak alkaline before plant cultivation, thus it is possible that application of elemental S at 15 days before sowing may have decreased soil pH. Lowered soil pH should have increased solubility of micronutrient, is this related to Cd tolerance?

We agree and changed the discussion accordingly.

Introduction of So to alkaline soils reduces soil pH through microbial oxidation [Yang et al.; Germida, et al. 1993]. This increases the solubility of micronutrient and becomes readily available to plant when compared to other S sources. However, the other S sources (ammonium sulfate, gypsum and magnesium sulfate) have a little or no considerable effect on the pH of soil. Moreover, in alkaline soils, free SO42- of ammonium sulfate, gypsum and magnesium sulfate can form complexes with Cd forming SO42--Cd complexes in Cd polluted soils [Gao et al.]. Therefore, SO42- in the form of ammonium sulfate, gypsum and magnesium sulfate become less available for plant uptake. Moreover, So ensures enough availability of SO42- by microbial action to be used by plant for longer times. This could be the reason for effectiveness of So when compared to other S sources. As of now, it is not possible to check the pH of soil, as the work has been already done.

I am also wondering if elemental S also can induce higher Cd tolerance for acid soils. The authors should discuss if elemental S is also effective on soils with different pH than that in the present study.

We agree and changed the discussion accordingly.

Notably soil pH is the most important parameter controlling mobility of cationic heavy metals in soils (Ervio, R. 1991). Lowering the soil pH increases the solubilization of Cd into the soil. Microbial oxidation of So introduced into the soil leads to a production of sulphuric acid. This process precipitate divalent cationic metals such as Cd (Tichy, R. et al. 1993; Germida, Jet al. 1993). So yes, So can be effective in soils with lower pH.

In conclusion (line 624) they describe that elemental S has a longer time of residence in soil. Is it related to higher Cd tolerance? I am wondering if their pots have holes on the bottom or not. If the pots have holes, sulfate ions form of ammonium sulfate, gypsum and magnesium sulfate would have been easily drained off while S in elemental sulfur remained longer. If there is no hole at the bottom of the pots, sulfate should not be drained off. They should discuss this possibility by describing about the presence of holes in the pots.

We agree and changed the discussion accordingly.

The SO42- from the ammonium sulfate, gypsum and magnesium sulfate are readily available for plant but their levels get depleted with the passage of time. Comparatively So amended soils have an edge over other S fertilizers since So is a slow S releasing fertilizer. The pots used in the experiments had holes, however the pots were watered as per the calculated field capacity, so that there would be minimum leaching.

If there are other possible mechanisms for the higher Cd tolerance by elemental S, they should describe in discussion.

We agree and changed the discussion accordingly.

Besides, So assisted microbial oxidation increases availability of other minerals such as Fe, N, Cu, Ca, Mn in soil has been reported (Kaya et al. 2009). Therefore, So supplement prevents the deficiency of mineral elements particularly in stressed conditions.

Followings are minor points.

Line 103: Describe the lengths of day and night.

Done

Line 108: (100S, lowS) (200S high S)

Actually, 100 mg S kg-1 soil is not low amount as S fertilizer. So, it is strange to write as “low”. Just 100S and 200S are enough.

Done

Lint 112: So

“0“ should be superscript.

Done

Table 1

Experiment 1, 2, and 3 are only described in this table. Show also in the texts in Materials and Methods which is Experiment 1, 2 and 3.

We added this information in paragraph 2.5.

Figure 1

Is this pie chart really tolerance index? There is not “means + SE” in the pie chart as in the legend. And “Giriraj had the highest tolerance index (0.789) (Line 248)” is not shown in this figure.

We deleted this chart and added a new line in Table 2.

Table 2

The unit for Cd level should not be “mg S kg -1 soil”. It should be “mg Cd kg -1 soil”.

Done

Table 2 and Table 4

Delete “g” or “e” from “nd” because it is impossible to perform statistical analysis with “nd”.

Right, we corrected this.

Figure 8 legend Line 477-478 Use “,” in place of “-”.

“under (A-E) control 0 mg Cd kg -1 soil + 0 mg S kg -1 soil, (B-F) 200 mg Cd kg -1 soil, (C-G) 200 mg S kg-1 soil, and (D-H) 200 mg Cd kg -1 soil + 200 mg S kg -1 soil.

should be

under (A, E) control 0 mg Cd kg-1 soil + 0 mg S kg-1 soil, (B, F) 200 mg Cd kg-1 soil, (C, G) 200 mg S kg-1 soil, and (D, H) 200 mg Cd kg-1 soil + 200 mg S kg-1 soil.

Done, thank you.

Line 523 more Cd in roots than shoots (Table 2). Table 2 should be Table 4.

Done

Round 2

Reviewer 2 Report

The authors revised well in discussion according to my previous comments. But there are still mistakes and points to be improved as follows.

Line 520 “S is immobile in plants”

Among essential elements, mobility of S is intermediate, less mobile compared to N, P, K, but mobile compared to Ca and most of micronutrients. Thus here, they have to write by comparing to other elements, for example,

“S is relatively immobile in plants compared to other macro nutrient such as N and K”

The authors answered to my previous comments as follows.

“The pots used in the experiments had holes, however the pots were watered as per the calculated field capacity, so that there would be minimum leaching.

They should include this in the Materials and method. It is important information if the pod has a hole or not, and how they watered.

The rightest line of Table 2 stuck out the page, thus I cannot read.

The authors still confuse “-“ and “,” in the legend of Figure 7. It should be as follows.

. (A-D) Cell viability test of roots cells and (E-H) leaf stomatal response of Brassica juncea cv. Girira under (A, E) control 0 mg Cd kg−1 soil + 0 mg S kg−1 soil, (B, F) 200 mg Cd kg−1 soil, (C,G) 200 mg S kg−1 soil, and (D, H) 200 mg Cd kg−1 soil + 200 mg S kg−1 soil. Bars (A, D) = 10 µM; bars (E, H) = 5 µM.

Author Response

Dear Professor,

thank you for your comments and suggestions. You will find here below a point-by-point list of answers to your concerns.

The authors revised well in discussion according to my previous comments. But there are still mistakes and points to be improved as follows.

Thank you for your appreciation.

Line 520 “S is immobile in plants”

Among essential elements, mobility of S is intermediate, less mobile compared to N, P, K, but mobile compared to Ca and most of micronutrients. Thus here, they have to write by comparing to other elements, for example, “S is relatively immobile in plants compared to other macro nutrient such as N and K”

We agree and changed the sentence accordingly.

The authors answered to my previous comments as follows.

“The pots used in the experiments had holes, however the pots were watered as per the calculated field capacity, so that there would be minimum leaching.

They should include this in the Materials and method. It is important information if the pod has a hole or not, and how they watered.

The rightest line of Table 2 stuck out the page, thus I cannot read.

We tried to reduce the font size. It should be OK now.

The authors still confuse “-“ and “,” in the legend of Figure 7. It should be as follows.

. (A-D) Cell viability test of roots cells and (E-H) leaf stomatal response of Brassica juncea cv. Girira under (A, E) control 0 mg Cd kg−1 soil + 0 mg S kg−1 soil, (B, F) 200 mg Cd kg−1 soil, (C,G) 200 mg S kg−1 soil, and (D, H) 200 mg Cd kg−1 soil + 200 mg S kg−1 soil. Bars (A, D) = 10 µM; bars (E, H) = 5 µM.

Right, we corrected the caption.

Kind regards and thanks again